# Antibiotic-Loaded Hyperbranched Polyester Embedded into Peptide-Enriched Silk Fibroin for the Treatment of Orthopedic or Dental Infections

**DOI:** 10.3390/nano12183182

**Published:** 2022-09-13

**Authors:** Zili Sideratou, Marco Biagiotti, Dimitris Tsiourvas, Katerina N. Panagiotaki, Marta V. Zucca, Giuliano Freddi, Arianna B. Lovati, Marta Bottagisio

**Affiliations:** 1Institute of Nanoscience and Nanotechnology, NCSR “Demokritos”, 15310 Aghia Paraskevi, Greece; 2Silk Biomaterials SRL, Via Cavour 2, 22074 Lomazzo, Italy; 3IRCCS Istituto Ortopedico Galeazzi, Cell and Tissue Engineering Laboratory, Via R. Galeazzi 4, 20161 Milan, Italy; 4IRCCS Istituto Ortopedico Galeazzi, Laboratory of Clinical Chemistry and Microbiology, Via R. Galeazzi 4, 20161 Milan, Italy

**Keywords:** carboxylated hyperbranched polyester, silk fibroin, antibiotics, implant-related infections, biofilm

## Abstract

The development of innovative osteoconductive matrices, which are enriched with antibiotic delivery nanosystems, has the invaluable potential to achieve both local contaminant eradication and the osseointegration of implanted devices. With the aim of producing safe, bioactive materials that have osteoconductive and antibacterial properties, novel, antibiotic-loaded, functionalized nanoparticles (AFN)—based on carboxylic acid functionalized hyperbranched aliphatic polyester (CHAP) that can be integrated into peptide-enriched silk fibroin (PSF) matrices with osteoconductive properties—were successfully synthesized. The obtained AFNPSF sponges were first physico-chemically characterized and then tested in vitro against eukaryotic cells and bacteria involved in orthopedic or oral infections. The biocompatibility and microbiological tests confirmed the promising characteristics of the AFN-PSF products for both orthopedic and dental applications. These preliminary results encourage the establishment of AFN-PSF-based preventative strategies in the fight against implant-related infections.

## 1. Introduction

The use of metal implants to restore the loss of bone function is a common and widespread procedure in both orthopedics and dentistry [1]. The main recipients of implant devices are middle-aged and elderly patients (>50 years old), which is a vulnerable population, frequently characterized by age-related comorbidities (i.e., hypertension, diabetes, hypothyroidism, etc.) and decreased biological activities (i.e., inflammatory, regenerative and remodeling phases) when facing tissue regeneration [2]. In particular, titanium, as a metal, is commonly used. The main advantages of titanium are its lack of allergenic or immunogenic potential, excellent corrosion resistance and osseointegration properties. However, despite the promotion of early cell adhesion and osseointegration, titanium implants also promote bacterial adhesion, leading to prosthetic infection. Thus, the failure of implants may be prompted by the presence of contaminants that are able to colonize the surface of medical devices, hampering the implant osseointegration and jeopardizing the clinical outcome of the surgical intervention. Indeed, in the presence of infections, osseointegration is erased by the relevant bone loss and soft tissue defects, resulting in poor implant stability and patient incapacitation. Implant-related infections are, indeed, the most feared complications of both orthopedic and dental surgery [3]. As soon as the medical device is implanted in the host, the ideal environment supporting the attachment of contaminants and the colonization of its abiotic surface is created. Once the microorganisms adhere to the implant surface, they secrete a thick, extracellular polymeric matrix to protect against immune system attacks and antibacterial treatments [4].

For all the aforementioned reasons, peri-implantitis and septic implant failures are intensively discussed topics that still require the development of preventative strategies to reduce the economic impact of revision procedures on national healthcare systems and to reduce the burden on invalid patients [3]. The use of biomaterials with bone-promoting supplements or growth factors, possibly loaded with drugs or antimicrobial agents, has been extensively investigated to enhance cell and matrix ingrowth and tissue function regeneration. Therefore, the development of novel biomimetic matrices has the invaluable potential to locally deliver antimicrobial agents and to support the osseointegration of implanted devices [3].

Silk fibroin has been widely used as a carrier to deliver drugs such as antibiotics, vaccines, insulin and other macromolecules. The mechanism behind the trapping and release of active molecules from silk has been discussed in detail in a number of studies [5,6,7,8,9,10], covering both soluble [10] and insoluble systems [11].

Many of silk fibroin’s (SF) peculiar features are very attractive in the field of drug delivery. It has excellent mechanical properties, high biocompatibility [12,13,14], interesting optical properties [15] and a suitable biodegradation pathway [16,17,18]. Moreover, it is potentially edible, nontoxic and relatively inexpensive.

Silk fibroin can be manipulated to obtain different physical forms, which offer a wide range of possibilities for the preparation of carriers. Most of these transformations can be achieved under mild conditions, without the need for organic solvents or harsh temperatures that could interfere with the bioactivity of encapsulated drugs [10,19]. This high flexibility in formulation makes SF an ideal candidate for drug delivery in many different districts.

Silk fibroin can also protect encapsulated drugs against external stress (e.g., temperature), and this property has proven very useful for delivering notoriously sensitive molecules, such as several antibiotics (e.g., doxycycline, rifampicin and erythromycin) [20,21].

Coming back to the orthopedic field, the use of silk fibroin seems to be very promising for the direct delivery of antibiotics, but also for the ability demonstrated by peptide-enriched silk fibroin (PSF) to stimulate apatite deposition in vitro and to accelerate the inward tissue deposition of native bone in vivo [22].

However, the indiscriminate use of antibiotics in conjunction with osteoconductive matrices can lead to toxicity issues in tissues or organs, especially at the doses necessary for infection eradication [23]. In addition, the use of ineffective materials, which are unable to locally deliver bactericidal drugs in a controlled manner, can induce the development of resistance mechanisms. Instead, antibiotic-loaded nanomaterials would be useful to enhance antimicrobial effects at low doses and to optimize the antibiotic long-term release, reducing the risk of the selection of resistant microorganisms [24,25]. Thus, functional dendritic nanocarriers with low toxicity, high loading capability and controlled release properties could be appropriate candidates [26,27,28,29]. Indeed, functional dendrimeric or hyperbranched polymers have been used to encapsulate or conjugate, through hydrolysable bonds, various antibiotic drugs in order to enhance their activity and reduce their severe side effects [30,31].

With the aim of producing safe bioactive materials with osteoconductive and antibacterial properties, the introduction of dendritic nanocarriers, encapsulating antimicrobial compounds in osteoconductive peptide-enriched silk fibroin matrices, was pursued. Previous studies have utilized a non-toxic PEGylated derivative of a hyperbranched aliphatic polyester for paclitaxel encapsulation and delivery [32], while a carboxylated derivative of the same polyester has been employed as a peptide delivery system [33].

This polyester is synthesized in the molten state, using 2,2-bis(hydroxylmethyl)propane-1,3-diol and 2,2- bis(hydroxymethyl) propionoic acid [34]. Due to its lipophilic interior, it was used to encapsulate lipophilic molecules. It was also found to be bio-degradable, since the enzymatic degradation of its ester bonds was observed [32]. Being water insoluble, it is advantageous for it to be appropriately functionalized with hydrophilic end groups (e.g., carboxylates or polyethylene glycol moieties), thus resulting in micellar-type nanosystems [32,33]. Additionally, after the initial screening of a number of hyperbranched polymers, the carboxylated derivative was found to be compatible with PSF solutions. Moreover, due to its anionic character, it can effectively interact with weakly basic antibiotics, forming nanoparticles in aqueous environments. The antibiotic-loaded functionalized nanoparticles (AFN) obtained were added to aqueous PSF solutions for the development of AFN-PSF sponges. The final AFN-PSF products were characterized and tested in vitro against a group of bacteria related to orthopedic or dental infections. The development of innovative osteoconductive matrices, enriched with antibiotic delivery nanosystems has the invaluable potential to achieve both the local delivery of antimicrobial agents and the osseointegration of implanted devices.

## 2. Materials and Methods

### 2.1. Chemicals and Reagents

Hyperbranched aliphatic polyester Boltorn^TM^ H40, BH40 (M_n_ = 5100 g mol^−1^, M_w_/M_n_ = 1.8, hydroxyl number = 485 mg KOH/g), bearing 44 hydroxyl end groups was kindly donated by Perstorp AB, Sweden. To remove low molecular weight fractions, BH40 was further fractionated by precipitation, following the procedure described in the literature [35]. High-purity solvents, such as pyridine, acetone, diethyl ether and dimethylformamide (DMF), were purchased from Merck KGaA (Calbiochem^®^, Darmstadt, Germany). Succinic anhydride, sodium bicarbonate (NaHCO_3_), lithium bromide (LiBr), dialysis tubes (M.W. cut-off: 1200), vancomycin hydrochloride (VC), minocycline hydrochloride (MC) and tetracycline base (TC) were purchased from Sigma-Aldrich Ltd. (Poole, UK), while phosphate buffer saline (PBS) was obtained from Biochrom (Berlin, Germany).

### 2.2. Synthesis and Physico-chemical Characterization of Carboxylated Hyperbranched Polyester

Partially carboxylated hyperbranched aliphatic polyester, with 20 carboxyl groups (CHAP), based on commercially available Boltorn^TM^ H40 (BH40), was prepared. To prepare this derivative, the starting BH40 polymer (0.2 mmol) was interacted with 4.6 mmol succinic anhydride in anhydrous pyridine (10 mL), at room temperature, for 48 h (Figure 1).

After the solvents evaporated, the residual was dissolved in acetone and the product was obtained through precipitation in diethyl ether. The raw product was dissolved in a 100 mM solution of NaHCO_3_ and dialyzed against water. The final product was received after lyophilization and its structure was confirmed by Fourier transform infrared (FTIR), ^1^H and ^13^C nuclear magnetic resonance (NMR) spectroscopies.

^1^H NMR (500 MHz, D_2_O): δ = 1.00–1.22 (m, CH_3_, CH_2_CH_3_), 2.30–2.45 (t, CH_2_CH_2_COO), 2.45–2.60 (t, CH_2_CH_2_COO^−^), 3.50–3.70 (m, OCH_2_), 4.01–4.45 (m, CH_2_COOCH_2_).

^13^C NMR (125.1 MHz, D_2_O): δ = 17.8 (CH_3_), 30.2 (CH_2_CH_2_COO), 31.9 (CH_2_CH_2_COO), 46.3 (CH_3_(OH)CCH_2_), 63.5–67.5 (CH2OH, CCH_2_O), 173–176 (ester CO), 180.5 (COO).

ATR-FTIR: wavelength (cm^−1^) = 3370 (ν, O-H), 2975 (ν_as_, CH_3_), 2940 (ν_as_, CH_2_), 2884 (ν_s_, CH_3_), 1726 (ν, C=O of ester groups), 1575 (ν_as_, COO^−^), 1397 (ν_s_, COO^−^), 1470 (δ_as_, CH_3_), 1458 (δ_s_, CH_2_), 1374 (δ_s_, C-CH_3_), 1122 (ν, C-O of ester groups), 1048 (ν, C-O of hydroxyl groups).

### 2.3. Encapsulation of Antibiotics in Carboxylated Hyperbranched Polyester Nanocarrier

Encapsulation of three different antibiotics, i.e., vancomycin, minocycline and tetracycline (Figure 2), in CHAP nanoparticles can be envisaged through electrostatic interaction between the positively charged amino groups of antibiotics and the anionic carboxylate groups of CHAP in water.

CHAP nanoparticles encapsulating vancomycin (CHAP_VC) were prepared by the drop-wise addition of 7.5 mL of VC aqueous solution (12 mg/mL), to 10 mL of CHAP aqueous solution (12 mg/mL), under vigorous stirring, and the pH was adjusted to 7.4. Under these conditions, nanoparticles were spontaneously formed and received after centrifugation at 20,000× *g*, they were repeatedly washed with water and were either lyophilized for FTIR studies, or re-dispersed in water for physico-chemical characterization and for the preparation of AFN-PSF sponges. CHAP nanoparticles encapsulating minocycline (CHAP_MC) were prepared by the drop-wise addition of 8.5 mL of MC aqueous solution (12.5 mg/mL) to 10 mL of CHAP aqueous solution (12 mg/mL), under vigorous stirring, and the pH was adjusted to 7.4. Spontaneously formed nanoparticles were received as described above. CHAP nanoparticles encapsulating tetracycline (CHAP_TC) were prepared by the drop-wise addition of 6 mL of TC methanol/water (1:1) solution (17 mg/mL), to 10 mL of CHAP aqueous solution (8.5 mg/mL), under vigorous stirring.

FTIR studies of the antibiotic-loaded nanoparticles were performed using a Nicolet 6700 spectrometer (Thermo Scientific, Waltham, MA, USA), equipped with an attenuated total reflectance (ATR) accessory, with a diamond crystal (Smart Orbit, Thermo Electron Corporation, Madison, WI, USA). The samples were firmly pressed against the diamond and the spectra were recorded at a resolution of 4 cm^−1^. A minimum of 64 scans were collected and signal averaged. Aqueous dispersions of antibiotic-loaded nanoparticles were characterized by UV–vis spectroscopy, dynamic light scattering (DLS) and zeta-potential. Specifically, the UV–vis spectra of the obtained nanoparticles dissolved in methanol solutions were recorded by employing a Cary 100 Conc UV–Visible spectrophotometer (Varian Inc., Mulgrave, Australia), to determine their loading capacity. Therefore, the corresponding calibration curves were established under the same experimental conditions. DLS studies of the aqueous nanoparticle dispersions were carried out by employing an AXIOS-150/EX (Triton Hellas, Thessaloniki, Greece) apparatus, with a 30 mW laser source and an Avalanche photodiode detector, at a 90° angle. For each sample, ten measurements were collected, and the autocorrelation function obtained was analyzed by employing the CONTIN algorithm [36], which is ideal for both monodisperse and polydisperse systems. DLS measurements were taken both immediately after the preparation of the nanoparticles and at various time intervals, for a total period of at least 2 days. Measurements of zeta potential were determined by a ZetaPlus (Brookhaven Instruments Corp, Long Island, NY, USA).

### 2.4. Preparation of AFN-PSF Sponges

Silk fibroin fiber cocoons were degummed in deionized water in the autoclave, at 120 °C, for 20 min, and extensively rinsed with warm and cold water to remove any sericin residues. Pure SF fibers thus obtained were dissolved in 9.3 M LiBr solution (Merck, Darmstadt, Germany), for 3 h, at 60 °C, under gentle agitation (SF concentration 10% *w*/*v*). The SF solution was then diluted 2:1 with distilled water, filtered, and dialyzed in a cellulose dialysis tube (cut off = 14,000; Sigma-Aldrich, Milan, Italy) until the complete removal of the salt was achieved. The final SF concentration was approximately 3.3% *w*/*v*.

Fibroin-derived anionic polypeptides (Cs) were prepared as previously described [37]. Briefly, an aqueous SF solution was treated with α-chymotrypsin (Sigma-Aldrich, Milan, Italy; enzyme-to-substrate weight ratio of 1:100), for 24 h, at 37 °C. The resulting precipitate was separated from the supernatant by centrifugation at 7400× g rpm for 20 min. The soluble anionic peptide fraction was recovered in the powder form from the supernatant by means of freeze-drying.

To prepare peptide-enriched silk fibroin (PSF) sponges, containing antibiotic-loaded functionalized CHAP nanoparticles (AFN), the prepared (as described above) SF aqueous solution was mixed with 10% w_Cs_/w_SF_ fibroin-derived anionic peptides, also dissolved in water (20 mg/mL), and the appropriate amount of AFNs dispersed in water. The resulting mixture was poured into a custom-made mold, frozen at −20 °C and finally freeze-dried. In order to standardize the dimensions of SF sponges and facilitate the set-up of experiments, custom-made molds of polydimethylsiloxane (PDMS, Farnell, Lainate, Italy) with cylindrical holes (8 mm Ø × 2 mm height) were prepared and used for the freeze-drying of AFN-PSF solutions.

According to this procedure, the following AFN-PSF sponges were prepared:SF+ 10% w_Cs_/w_SF_ Cs peptides + 10% or 20% w_AFNs_/w_SF_ CHAP_VC (PSF@CHAP_VC10 or PSF@CHAP_VC20, respectively).SF+ 10% w_Cs_/w_SF_ Cs peptides + 10% or 20% w_AFNs_/w_SF_ CHAP_MC (PSF@CHAP_MC10 or PSF@CHAP_MC20, respectively).SF+ 10% w_Cs_/w_SF_ Cs peptides + 10% or 20% w_AFNs_/w_SF_ CHAP_TC (PSF@CHAP_TC10 or PSF@CHAP_TC20, respectively).

In addition, PSF sponges containing 10% or 20% w_CHAP_/w_SF_ CHAP (PSF@CHAP10 or PSF@CHAP20) were also prepared. Additionally, it should be noted that the direct addition of antibiotics to PSF solution induced an instantaneous coagulation of the protein, thus making it impossible to create useful sponges that could also be used for comparison.

### 2.5. In Vitro Release of Antibiotics from AFN-PSF Sponges

The evaluation of the antibiotic release from AFN-PSF sponges over time was carried out by immersing 100 mg of antibiotic-loaded sponges in 30 mL of PBS (pH = 7.4) and incubating them at 37 °C, in an orbital shaker set at 200 rpm (SI500, Stuart, UK). After each predetermined time interval, 1 mL of eluate was withdrawn from the released medium and 1 mL of fresh buffer was added in order to keep the release medium volume constant. The concentration of antibiotics was then measured using a Cary 100 Conc UV-Visible spectrophotometer (Varian Inc.), at 370 nm for TC, at 356 nm for MC and at 280 nm for VC. Therefore, a standardized absorbance/concentration calibration curve for each of the tested compounds in PBS was also determined. The cumulative fraction of antibiotic release over time was given by the following Equation (1):drug release (%) = (*D_t_/D_tot_)* × 100(1)
where *D_t_* is the amount of antibiotic released at time *t*, while *D_tot_* is the total antibiotic present in AFN-PSF sponges. *D_tot_* was separately determined by dissolving 15 mg of each AFN-PSF sponge sample in 1 mL of an aqueous solution of 9 M LiBr, to achieve complete solubilization of the sample. The antibiotic content of each sample was then determined by the addition of 50 μL of this solution to 1 mL of PBS and registering the UV-vis spectrum.

### 2.6. Cell Toxicity Studies of PSF Sponges Loaded with 10% or 20% CHAP

The biocompatibility of PSF@CHAP10 and PSF@CHAP20 sponges was initially studied before continuing with microbiological analyses to rule out any toxic effects caused by the combination of SF, Cs peptides and CHAP.

Briefly, NIH-3T3 murine fibroblasts (ATCC^®^ CRL-1658) were seeded in 6-well plates, at a density of 3 × 10^5^ cells/cm^2^ and cultured in complete medium (CM), consisting of Dulbecco’s Modified Eagle’s Medium (DMEM, ATCC^®^ 30-2002), 10% calf bovine serum (ATCC^®^ 30-2030™), 100 U/mL penicillin-streptomycin, 2 mM L-glutamine, 1% sodium pyruvate, and 1% HEPES (all from Gibco, Life Technologies, Carlsbad, CA, USA). PSF@CHAP10 or PSF@CHAP20 sponges were placed in Transwell (Corning) for an indirect co-culture with the underlying cell monolayer. Three sponges (mean weight: PSF@CHAP10 2.17 ± 0.73 mg; and PSF@CHAP20 2.25 ± 0.91 mg) were placed in each insert to keep a ratio of 1 sponge per mL. NIH-3T3 cells cultured in CM and supplemented with 0.1% Triton-X100 (Sigma-Aldrich) served as the positive control (PC), and cells cultured in fresh CM were used as the negative control (NC).

After a 24 h indirect culture, cell viability was assessed by the MTT assay, in which the reduction of yellow tetrazolium salt 3-(4,5-dimethylthiazole-2-yl)-2,5-diphenyltetrazolium-bromide (MTT, Sigma-Aldrich, St. Louis, MI, USA) to purple formazan is carried out by viable and metabolically active cells. Briefly, cell monolayers were washed three times with PBS (Gibco, Life Technologies, Carlsbad, CA, USA), then 1 mL of MTT in DMEM, without phenol red (0.5 mg MTT/mL), was added to each well and incubated at 37 °C and 5% CO_2_, for 2 h. Thereafter, the MTT solution was gently removed, and the formazan crystals were solubilized with 500 μL of a 1:10 solution of hydrochloric acid, 1N in isopropanol (both Sigma-Aldrich, St. Louis, MI, USA). The absorbance was read at 570 nm by means of a microplate reader (Victor X3, Perkin Elmer, Waltham, MA, USA). Three replicates were considered per sample and the data were reported as absorbance values.

### 2.7. Microbiological Analyses: Bacterial Strains and AFN-PSF Products

The clinical strains used to test AFN-PSF products were isolated from implant-related infections at the Laboratory of Clinical Chemistry and Microbiology, of the IRCCS Galeazzi Orthopedic Institute (Milan, Italy). In particular, two biofilm-producer strains of *Staphylococcus aureus* (a methicillin-resistant strain, MRSA, and a glycopeptide-intermediate strain, GISA), and *Staphylococcus epidermidis* (a methicillin-resistant strain, MRSE, and a glycopeptide-intermediate strain, GISE) were selected to evaluate both the antibacterial activity of AFN-PSF sponges for orthopedic applications and the development of antibiotic resistances. In particular, the PSF@CHAP_VC10 and PSF@CHAP_VC20 sponges, as well as the non-drug-loaded PSF@CHAP10 and PSF@CHAP20 sponges were tested.

Similarly, for dental applications, the reference strains *Aggregatibacter actinomycetemcomitans* (ATCC^®^ 29522), *Streptococcus mutans* (ATCC^®^ 25175), and two clinical isolates, one *Escherichia coli*, and one *Enterococcus faecalis*, were selected to test the PSF@CHAP_MC10, PSF@CHAP_MC20 and PSF@CHAP_TC10 sponges, as well as the non-drug-loaded PSF@CHAP10 and PSF@CHAP20 sponges.

### 2.8. Evaluation of the Antibacterial Activity of AFN-PSF Sponges

To test the antibacterial characteristics of AFN-PSF sponges and the non-antibiotic- loaded ones, time-killing curves were determined. Briefly, 100 μL of a bacterial suspension, containing approximately 1.5 × 10^6^ CFU/mL, was inoculated into Eppendorf tubes containing 900 μL of brain–heart infusion (BHI, Sigma Aldrich, St. Louis, MI, USA) broth, and one sponge was added, according to the tested formulation.

A bacterial suspension in BHI, without the addition of any sponge, was used as the positive control (CTRL+). Microbial counts were performed after 0 (T0), 24 (T1), 48 (T2), 72 (T3), 96 (T4) and 168 (T7) hours of incubation by serially diluting 10 μL of bacterial suspensions and by drop plating 10 μL of the proper dilution on Tryptic Soy agar (TSA, Sigma Aldrich) plates [38]. After 24 h of incubation at 37 °C, viable colonies were counted, and the number recorded. The results were expressed as log_10_ (CFU/mL). The bactericidal activity was defined as a 3-log reduction in CFU/mL (99.9% kill) from the initial inoculum.

### 2.9. Evaluation of the Selection for Bacterial Resistance after Multi-Step Exposure to ANF-PSF Sponges

The ability of the antibiotic-loaded silk fibroin products to select the tested bacterial strains for antibiotic resistance was investigated. Briefly, the bacterial strains from an overnight culture were suspended in sterile saline (Fresenius Kabi) to a turbidity of 0.5 McFarland (approximately 1.5 × 10^8^ CFU/mL) and plated all over the Petri dish surface with a sterile cotton swab. MRSA, GISA, MRSE and GISE were seeded onto Mannitol Salt agar (MSA, Sigma Aldrich); *E. coli* onto MacConkey (MAC, Oxoid, Hampshire, UK) agar; *E. faecalis* onto TSA; *A. actinomycetemcomitans* onto Schaedler Agar, with 5% sheep blood (SCH, Thermo Fisher Scientific, Waltham, MA, USA); and *S. mutans* onto Mitis Salivarius agar (M/S agar, Merck, Kenilworth, NJ, USA).

Thereafter, one AFN-PSF sponge was placed in the middle of the agar plate and incubated for 24 h, at 37 °C. The following day, the zone of inhibition (ZOI)—the area without any visible bacterial growth—was measured in the nearest whole millimeters, as judged by the unaided eye and the result recorded. Bacteria grown on the edge of the ZOI were suspended in sterile saline, to a turbidity of 0.5 McFarland, and seeded on another fresh agar plate, on which the AFN-PSF sponge, removed from the previous plate, was placed. This process was repeated for 7 days (from T1 to T7). At T7, bacteria were either suspended in BHI broth with 10% glycerol, then stored at −80 °C until further analysis or plated on fresh agar plates without any sponge for another 7 consecutive days of culture (from T8 to T14). This last step aimed to evaluate whether the potential increase in antibiotic resistance at T7 was stable or transient. At T14, the bacteria were frozen at −80 °C, as reported above.

### 2.10. Determination of Minimum Inhibitory Concentration

To assess whether the bacteria were selected for antibiotic resistance, the minimum inhibitory concentration (MIC) was determined at T0, T7 and T14 by the broth microdilution method, according to the European Committee on Antimicrobial Susceptibility Testing (EUCAST) guidelines [39].

A microbial suspension was prepared for each bacterial strain in Muller Hinton (MH) or BHI broth, at an optical density equal to 0.5 McFarland (1.5 × 10^8^ CFU/mL). Thereafter, 10 μL of 5 × 10^6^ CFU/mL suspension was inoculated in a 96-well microplate, containing 90 μL of a serial 2-fold dilution of either vancomycin, minocycline or tetracycline, according to the previously tested experimental conditions. MIC values were measured after 24 h of incubation, at 37 °C. A significant decrease in susceptibility was defined as >4-fold increase in MIC, in relation to the pre-exposure MIC values [40].

### 2.11. Statistical Analysis

Three biological and three technical replicates were carried out to evaluate the antibacterial activity of AFN-PSF products, by means of time-killing curves. Two biological and three technical replicates for each condition were carried out to assess the development of antimicrobial resistance.

The normal distribution of the data was determined through the Shapiro–Wilk test. Comparisons among groups were performed by means of Kruskal–Wallis test, coupled with Dunn’s post hoc test (GraphPad Prism v5.00 Software, Palo Alto, CA, USA). All data are expressed as means ± standard deviation (SD), unless otherwise specified. Values of *p* < 0.05 were considered statistically significant.

## 3. Results

### 3.1. Synthesis and Characterization of Carboxylated Hyperbranched Polyester (CHAP)

The partial functionalization of the commercially available hyperbranched aliphatic polyester Boltron H40^TM^ (BH40) was achieved by the reaction of the hydroxyl end groups of BH40 (44 OH groups per molecule [32]) with succinic anhydride, in anhydrous basic conditions (pyridine). The introduction of the carboxylic groups was established by ^1^H NMR and ^13^C NMR (Appendix A). Specifically, the substitution of BH40 was confirmed by the appearance in the ^1^H NMR spectrum of the characteristic peaks at ~2.50 and ~2.40 ppm, attributed to the α- and β-methylene protons adjacent to the carboxylic group, respectively, as well as the peak at 180.5 ppm in the ^13^C NMR spectrum, which was attributed to the carbon of newly introduced carboxylate end groups. The degree of substitution was determined by the integration of the signals in the ^1^H NMR spectrum at 1.00–1.20 ppm, attributed to the protons of methyl and the methylene groups of BH40, and the peaks at 2.50 and 2.40 ppm, attributed to α- and β -CH_2_, relative to the carboxylic groups, respectively. An average of 20 carboxylate groups were found to be attached to hyperbranched polyester, suggesting that ca. 45% of the hydroxyl end groups of BH40 were substituted. Moreover, the FTIR spectrum of CHAP (Appendix A) reveals, in addition to the bands of BH40, the presence of two new bands at 1575 cm^−1^ (strong) and 1398 cm^−1^, attributed to the symmetric and anti-symmetric vibrations of the carboxylic group that, further to the NMR results, denotes that the carboxylate end groups of the polymer were negatively charged due to the addition of NaHCO_3_ during its workup.

### 3.2. Development of Antibiotic-Loaded CHAP Nanoparticles

The carboxylate derivative of hyper-branched aliphatic polyester (CHAP) was used to encapsulate vancomycin, tetracycline, gentamicin and minocycline (Figure 2). Due to the presence of positively charged amino groups in the antibiotics, a strong interaction with the highly carboxylated polymer was expected through ionic bonds, without, however, excluding the possibility of the formation of hydrogen bonds due to the presence of electron donor and electron acceptor groups in the interacting compounds. Various conditions were studied, which finally led to the establishing of the protocols that result in the formation of nanoparticles that have high dispersion stability in water and sizes in the range of 100 nm. Thus, the average diameter of the resulting CHAP_VC, as determined by DLS, was ~80 nm and their zeta potential was −39.5 mV; however, the drug loading was found to be 39.5% *w*/*w*. CHAP_MC have an average diameter size of 110 nm, a zeta-potential value of −50.0 mV and a drug loading of 45.5% *w*/*w*; however, CHAP_TC nanoparticles have an average diameter size of 90 nm, a zeta potential value of −52.5 mV and a drug loading of 49 % *w*/*w*. The zeta potential values obtained were lower than that of CHAP (−58.0 mV) due to the encapsulation of positively charged antibiotics, but they were still negative, which renders them stable in aqueous media due to the electrostatic repulsion of the negatively charged nanoparticles.

FTIR studies were carried out to examine the nature of the interaction between CHAP and the antibiotics. The spectra of the drug-loaded CHAP nanoparticles were essentially the result of the addition of the two separate spectra, although several characteristic vibration modes were registered as shifted, when compared to their original spectra. In the spectrum of CHAP_TC, both the bands of pure CHAP and TC were evident, as expected. However, some variations in the positions of several characteristic bands were evident (Appendix A). Specifically, the peak of CHAP at 1726 cm^−1^ shifted to 1735 cm^−1^, suggesting the formation of an H-bond of the carbonyl groups with OH or NH_2_ groups of TC. On the other hand, the band at 1646 cm^−1^ of TC—attributed to both the C=O quinone groups and the Amide I band of its primary amide group—shifted in the spectrum of CHAP_TC, to 1657 cm^−1^. Finally, the skeletal, aromatic ring breathing modes of the quinone groups of TC, originally present at 1518 cm^−1^ and 1452 cm^−1^, shifted to 1511 cm^−1^ and 1444 cm^−1^, thus, suggesting their participation in a hydrogen-bonded network. Indeed, it is known that unsaturated groups, such as the aromatic C=C groups in this case, can also act as proton acceptors in H-bonds [41].

Upon the encapsulation of minocycline CHAP_MC nanoparticles, the characteristic band of MC at 1649 cm^−1^—attributed to both the C=O quinone groups and the Amide I band of its primary amide group—shifted to 1653 cm^−1^ and became broader in the spectrum of CHAP_MC, thus, suggesting the participation of these groups in an intermolecular hydrogen bonding network. A substantial reduction in the intensity of the 1575 cm^−1^ and 1398 cm^−1^ bands, typical of ionized carboxylic groups (COO^-^), and the appearance of a new low intensity band at 1652 cm^−1^ was indicative of neutral acid groups (COOH) also participating in the internal hydrogen bond formation (Appendix A) [42]. These observations also support the hypothesis that the interaction between the two components of this system is not only of the ionic, acid–base type, but that it also involves hydrogen bonding interactions.

In the spectrum of CHAP_VC nanoparticles, both the bands of pure CHAP and VC were evident, with some alterations, in the positions of several typical bands of both compounds. Specifically, the peak of CHAP at 1726 cm^−1^ shifted to 1734 cm^−1^, suggesting the formation of an H-bond of the carbonyl groups with the OH, amide or NH_2_ groups of VC. On the other hand, the band at 1645 cm^−1^ of VC—attributed to the Amide I band of its amide group—in the spectrum of CHAP_VC, shifted to 1658 cm^−1^, while the shoulder at 1674 cm^−1^ and the absence of the 1575 cm^−1^ band (Appendix A) suggest, as before, that in CHAP_VC the carboxylic groups of CHAP are no longer ionized but are non-ionic and participate in the internal hydrogen bond formation with VC.

### 3.3. Characterization of PSF and AFN-PSF Sponges

In the FTIR spectrum (Appendix A) of Cs containing SF sponges, the PSF (Cs 10%) displays the two dominant bands, at 1622 cm^−1^ and 1513 cm^−1^, assigned to Amide I and II vibrations, as well as the band at 1264 cm^−1^, attributed to the Amide III vibration of the crystalline *β*-sheet structure of the protein [43]. The bands that appear as shoulders of the previous amide bands, at 1660 cm^−1^ and at 1540 cm^−1^, as well as the bands at 1230 cm^−1^ and 1066 cm^−1^, were assigned to the Amide I, II, III and IV bands of the peptide, respectively, in a random coil conformation [44]. The intensity ratio of the two Amide III components at 1231 cm^−1^ and at 1264 cm^−1^ was used to provide an estimate of the crystallinity of the SF. In our case, the obtained ratio of 0.56 is close to that of native silk fibroin [45]. The FTIR spectrum of the PSF@CHAP20 sponges was mainly characterized by the presence of the absorption bands of PSF, while, due to the low blending ratio employed (20%), only the high intensity bands of CHAP, at ca. 1730 cm^−1^ (C=O stretching), at 1123 cm^−1^ (C-O stretching of ester groups) and at 1055 cm^−1^ (C-O stretching of the hydroxyl groups) were evident in the PSF@CHAP20 spectrum as low intensity bands (Appendix A). Moreover, the complete absence of the strong, 1575 cm^−1^ band of CHAP and the concomitant appearance of a shoulder at approximately 1650 cm^−1^ was observed in the spectrum of the PSF@CHAP20.

### 3.4. In Vitro Release of Antibiotics from AFN-PFS Matrices

The in vitro antibiotic release from AFN-PSF sponges was evaluated in PBS at 37 °C for a three-day period. The release profiles of the antibiotics from AFN-PSF sponges are shown in Figure 3.

In the case of PSF@CHAP_TC, 65% of the TC was released from the corresponding sponges during the first 30 min, followed by a slow release of approximately 15%, which reached a plateau for the next 3 days. In contrast, in the case of PSF@CHAP_MC, the corresponding sponges released close to 100% of the MC for the first 2 h. It is interesting to note that in the following time period (1–3 days) a considerable decrease in the MC release percentage was observed, which is attributed to the degradation of MC in the aqueous media of neutral pH, in line with the literature’s data [46]. On the other hand, in the case of PSF@CHAP_VC, only 55% of the VC was released from the corresponding sponges, with a steady rate during the first 1 h, followed by a significant slow release of ~30% over the next 3 days. It is interesting to note that when the release profiles were plotted on a logarithmic time scale, in all cases the antibiotic release increased during the first 2 h, in an almost linear fashion with respect to the logarithmic time (Figure 3, insert). This behavior was attributed to the release mechanism controlled both by diffusion and polymer (i.e., PSF) swelling [47].

### 3.5. Cytotoxicity Evaluation of PSF Sponges, Loaded with 10% or 20% CHAP

The potential cytotoxic effect of CHAP was tested on NIH-3T3 murine fibroblasts, indirectly cultured in the presence of PSF sponges, loaded with 10% or 20% CHAP. The results obtained by the MTT assay are reported in Figure 4.

The absorbance values of the NIH-3T3 cells cultured in fresh CM (NC) were significantly higher than the PC for *p* < 0.001, as expected. Similarly, the indirect culture of fibroblasts in the presence of either PSF@CHAP10 or PSF@CHAP20 sponges showed higher absorbance compared to the PC (*p* < 0.01), denoting a regular cell proliferation and biocompatibility of the tested products.

### 3.6. Antibacterial Activity of AFN-PSF Sponges

The antibacterial capability of AFN-PSF sponges was tested by killing curves over time.

Regarding the orthopedic infection-related strain bacteria, AFN-PSF sponges, enriched with 20% vancomycin-loaded CHAP nanoparticles (PSF@CHAP_VC20), killed all tested bacteria after 48 h (T2) of culture (Figure 5). A slower bacterial death occurred in the presence of PSF@CHAP_VC10, where MRSA, MRSE and GISA were eradicated only after 72 h (T3). Otherwise, GISE showed a higher susceptibility to PSF@CHAP_VC10, given that a >99.9% reduction in the colony counts was already observed after 48 h (T2). The viability of all the tested clinical isolates was not affected by the presence of either the PSF@CHAP10 or the PSF@CHAP20 sponges. Indeed, the growth curves of the staphylococci cultured in the presence of these unloaded sponges were similar to that of the bacteria grown in the BHI (Figure 5).

Since a 3-log decrease in the number of CFU per mL, compared to the initial inoculum, was detected over time, PSF@CHAP_VC10 and PSF@CHAP_VC20 can be defined as bactericidal under the current experimental conditions.

On the other hand, among the AFN-PSF sponges, PSF@CHAP_MC20 exhibited the greatest bactericidal activity against all tested bacterial strains related to dental infections (Figure 6). Specifically, this formulation was already effective against *A. actinomycetemcomitans* and *S. mutans* after 24 h (T1), whereas the death of *E. coli* and *E. faecalis* cells occurred after 48 h (T2). Additionally, among the tested bacterial strains, the most susceptible strain was *A. actinomycetemcomitans*, with more than 99.9% cell mortality already detected after 24 h (T1), when cultured in the presence of PSF@CHAP_TC10 and PSF@CHAP_MC20. However, the amount of minocycline released from the PSF@CHAP_MC10 sponges was not sufficient to kill bacteria at T1 (Figure 6), but only after 48 h (T2). For all strains tested, other than *A. actinomycetemcomitans*, a slower bactericidal activity of PSF@CHAP_TC10 was observed at T3.

Finally, similar to the results obtained for the AFN-PSF formulations for orthopedic use, the viability of the tested oral bacteria was not negatively affected by the presence of PSF@CHAP10 or PSF@CHAP20 sponges, their growth curve being similar to that of cells in BHI (Figure 6). A physiological decrease in the viability of *A. actinomycetemcomitans* and *S. mutans* was observed, starting from T2. The differences displayed in *A. actinomycetemcomitans* viability of PSF@CHAP10 and PSF@CHAP20 in the present study, were not statistically significant compared to BHI.

### 3.7. Evaluation of the Selection for Bacterial Resistance after Multi-Step Exposure to AFN-PSF Sponges

Bacteria were cultured in the presence of the same AFN-PSF sponge, during a 7-day period, to evaluate whether the antibiotic release might induce the development of any transient or stable resistance trait.

All the staphylococcal strains tested showed a constant decrease in ZOI over time. In particular, PSF@CHAP_VC20 was responsible for the larger (but not statistically significant) halos that were free of bacteria, compared to PSF@CHAP_VC10 (Figure 7).

The lack of a stable or transitory adaptation was then confirmed by a MIC analysis of all the clinical isolates exposed to the vancomycin released by the corresponding AFN-PSF sponges. A slight increase in the MIC values was found for the MRSE strain, although this was not significant (Table 1). Indeed, the MIC variations recorded never exceeded the 4-fold dilution, so that no significant change in the susceptibility profile was observed after a multi-step exposure to AFN-PSF sponges.

Under the experimental conditions tested, no adaptation was observed after the exposure of oral bacteria to the PSF@CHAP_MC10, PSF@CHAP_MC20 or PSF@CHAP_TC10 sponges. The tetracycline release from PSF@CHAP_TC10 was accelerated, compared to both PSF@CHAP_MC10 and PSF@CHAP_MC20, which released minocycline more consistently over time (Figure 8).

The absence of antimicrobial resistant traits was confirmed by the determination of the MIC values. Indeed, no significant change in the susceptibility profile was observed in the bacteria when exposed to AFN-PSF products for dental applications (Table 2).

## 4. Discussion

Implant-related infections are a major burden for orthopedics and dental surgery. The formation of a bacterial biofilm is one of the main causes of these infections, which have serious consequences, such as the loosening of implants. Current antibiotic prophylaxis/therapy is inadequate to prevent the formation of a biofilm and gives rise to antibiotic resistance. Thus, the development of novel, biomimetic antibiotic-enriched matrices has the invaluable potential to achieve both the local delivery of antimicrobial agents and the osseointegration of implanted devices. This study aimed to produce safe bioactive materials with osteoconductive and antibacterial properties, by using antibiotic-loaded functionalized nanoparticles (AFN) in osteoconductive peptide-enriched silk fibroin matrices (PSF), to be employed both as drug delivery and regenerative materials.

While the process that leads to PSF is very well consolidated [22], their combination with AFN made from hyperbranched polymers required an optimization effort. Silk fibroin dissolved in aqueous solutions, prepared as described above, has a negative overall zeta potential (in the order of −10 mV) [48], primarily because the repetitive part of the protein entails hydrophobic domains with intervening short, negatively charged hydrophilic blocks [49]. This requires the use of neutral or negative hyperbranched polymers or nanoparticles in order to avoid electrostatic interactions that would destabilize the silk fibroin solution, resulting in precipitation. Initial testing with a series of positively charged hyperbranched polyethyleneimines, functionalized or not, corroborated this prediction by leading, upon their addition to SF or PSF solutions, to severe coagulation. This was also observed when the antibiotics studied in this work, which also bear amino groups, were added in the above solutions. On the other hand, the non-toxic hyperbranched aliphatic polyester (BH40) or its derivatives—previously employed as drug delivery systems [32,33]—proved to be compatible with SF or PSF solutions, and also afforded, after lyophilization, polymer-PSF sponges, with non-deteriorated mechanical properties. Therefore, in this study, the carboxylate functionalized hyperbranched aliphatic polyester (CHAP) was further utilized due to its compatibility with PSF solutions. Moreover, due to its carboxylic acid moieties, it was able to interact and encapsulate the antibiotics, forming nanoparticles. The interaction of CHAP with the antibiotics in water was elaborated by modifying the concentrations of the interacting solutions—the CHAP: antibiotic ratio—and by carefully monitoring the pH values of the resulting solution in order to obtain stable nanoparticle dispersions, with average hydrodynamic values of approximately 100 nm and negative zeta potential values. The latter is essential to obtain stable nanoparticle dispersions due to electrostatic repulsions, and to prevent coagulation in the next step of mixing with PSF solutions.

The nature of the interaction of CHAP with vancomycin, minocycline and tetracycline was examined using infrared spectroscopy. Their interaction was found to be mainly electrostatic in nature, but hydrogen bond interactions were also involved. Specifically, the spectra provide evidence of H-bond formation between the carbonyl groups of CHAP and the hydroxyl, amino and amide groups of TC or MC. While, in the case of CHAP_VC, the carboxylic groups of CHAP are non-ionized and participate, together with the carbonyl groups of CHAP, in the formation of H-bonds with the hydroxyl, amide or amino groups of VC. Finally, the infrared spectra of the PSF@CHAP sponges provide evidence that the carboxylic groups of CHAP form hydrogen bonds with complementary PSF groups, which might account for the observed homogeneous solution and the mechanical stability of the resulting freeze-dried sponges. Furthermore, the PSF sponges were also characterized by FTIR spectroscopy, revealing that the crystallinity of SF is analogous with that of native silk fibroin [45], as estimated by the intensity ratio of the two Amide III components, at 1231 cm^−1^ and at 1264 cm^−1^. Moreover, due to the complete absence of the 1575 cm^−1^ band of CHAP and the presence of a shoulder at approximately 1650 cm^−1^, it is obvious that the carboxylic groups are no longer anionic. In addition, the observed carbonyl frequency change suggests that carboxylic groups form hydrogen bonds [42] with the complementary PSF groups. Finally, the spectra of the PSF sponges with antibiotic-loaded CHAP (10 or 20% loading) were, in all cases, a simple addition of both components, without displaying any distinct variations in band locations or intensities.

The success of the synthesis of the PSF sponges enriched with functionalized, antibiotic-loaded nanoparticles made it necessary to test the biocompatibility and the antibacterial activity of the products that were developed.

The evaluation of any potential cytotoxic adverse effects on the eukaryotic cells was performed as a first step, since the failure of this experimental phase would have precluded the subsequent microbiological investigations. Indeed, the major concern was the potential cytotoxic effects of any chemical residue derived from the synthesizing process. According to the ISO standards [50], https://www.iso.org/standard/36406.html, (15 June 2009)] for the biological evaluation of medical devices, cell viability <70% compared to the NC has to be interpreted as a cytotoxic potential. The results obtained in this study through the MTT viability test showed good cytocompatibility features in both PSF@CHAP10 and PSF@CHAP20, with 98.7 and 97.1 viability percentages, respectively. Therefore, the microbiological evaluations were conducted, based on the assessment of the antibacterial capacity of AFN-PSF sponges, using kill curves over time. Even though the concentration of antibiotics encapsulated in AFN-PSF sponges was largely sufficient to kill bacteria, according to their MIC values, it was important to determine whether the controlled release of antibiotic molecules over time was enough to complete the bacterial eradication at early time points. Again, this experimental assessment yielded positive and promising results for the use of AFN-PSF sponges as potent orthopedic and dental products. In particular, the higher concentration of vancomycin loaded within the PSF@CHAP_VC20 orthopedic products resulted in a faster bacterial clearance than the PSF@CHAP_VC10 under all tested conditions. It was easy to anticipate that the antibiotic concentration is directly proportional to the antimicrobial effect, as was also demonstrated by the data obtained by testing PSF@CHAP_MC10 and PSF@CHAP_MC20 against the dental bacterial panel. Unfortunately, PSF@CHAP_TC20 was unable to test for antibacterial activity. Indeed, the synthesis process of AFN-PSF sponges, enriched with 20% CHAP_TC, generated products too brittle to be easily manipulated during experimental procedures. Due to the fragility of the PSF@CHAP_TC20 sponges, which resulted in collapse during the tests, the data obtained in the evaluation of these products were not analyzed and presented in this study.

Finally, considering the promising results obtained from the cytotoxicity and antimicrobial tests, the last important issue to exclude was the accidental selection of antibiotic- resistance traits in bacteria, due to the selective pressure of the antibiotic released by AFN-PSF sponges in a controlled, time-dependent manner. To accomplish that, both orthopedic and dental-related bacteria were cultured in the presence of the AFN-PSF sponges, over a 7-day period and, subsequently, were subcultured for another 7 days to exclude any transient or stable reduced susceptibility to antibiotics. None of the AFN-PSF products produced increased the MIC by four times the pre-exposure MIC values, in any of the bacteria tested.

Based on our findings, PSF@CHAP_VC20 and PSF@CHAP_MC20 were the safest and most effective products for the eradication of in vitro bacterial contamination. Although promising, the results described in the current work need to be further investigated to reinforce the clinical value of the products with more scientific evidence. Indeed, one of the limitations of this study may be the use of a relatively low number of clinical isolates. There will be the need to broaden future analyses, including the use of different bacterial species and antibiotic susceptibility profiles, in order to allow the use of AFN-PSF sponges in the management of both orthopedic and dental implant-related infections.

## 5. Conclusions

With the aim of producing safe bioactive materials with osteoconductive and antibacterial properties, peptide-enriched silk fibroin (PSF) sponges, enriched with antibiotic-loaded functionalized nanoparticles (AFN), were successfully synthesized and described in the present study. The biocompatibility and microbiological tests confirm the promising characteristics of the product PSF@CHAP_VC20, designed and developed for orthopedic use and of PSF@CHAP_MC20, meant for dental applications. Further evaluations will be performed in the future on a wider range of pathogens in order to expand the use of the products as preventative strategies in the fight against implant-related infections.

## Figures and Tables

**Figure 1 nanomaterials-12-03182-f001:**
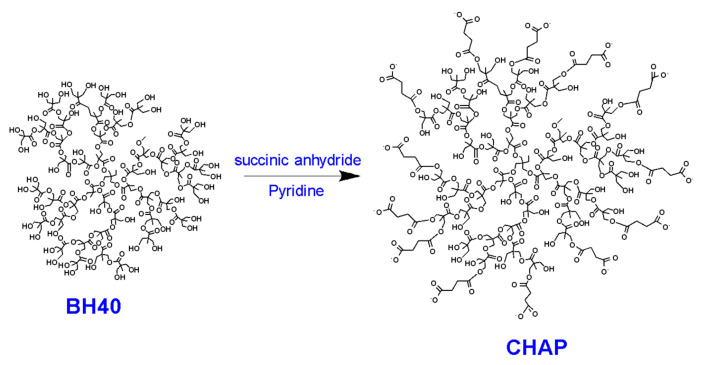
Reaction scheme for the preparation of CHAP.

**Figure 2 nanomaterials-12-03182-f002:**
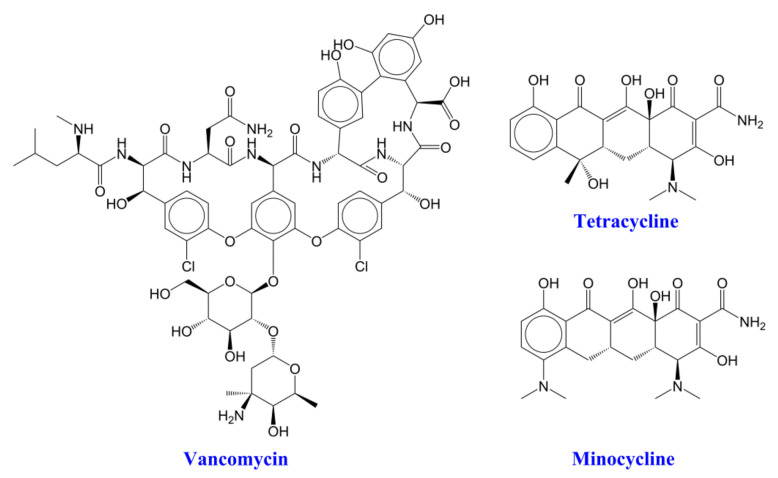
Chemical structures of the selected antibiotics.

**Figure 3 nanomaterials-12-03182-f003:**
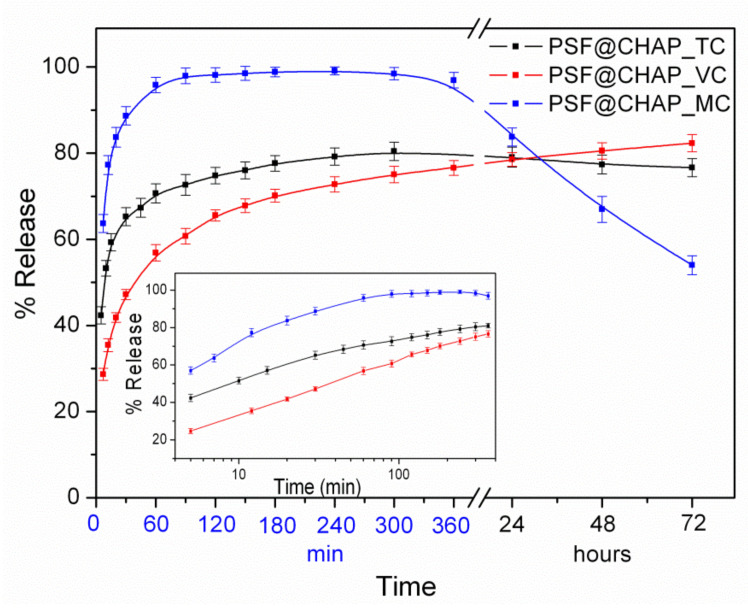
Release profiles of the percentages of the total antibiotics in PBS, at 37 °C, from the corresponding AFN-PSF sponges. In the insert, the corresponding release profiles are presented on a logarithmic time scale.

**Figure 4 nanomaterials-12-03182-f004:**
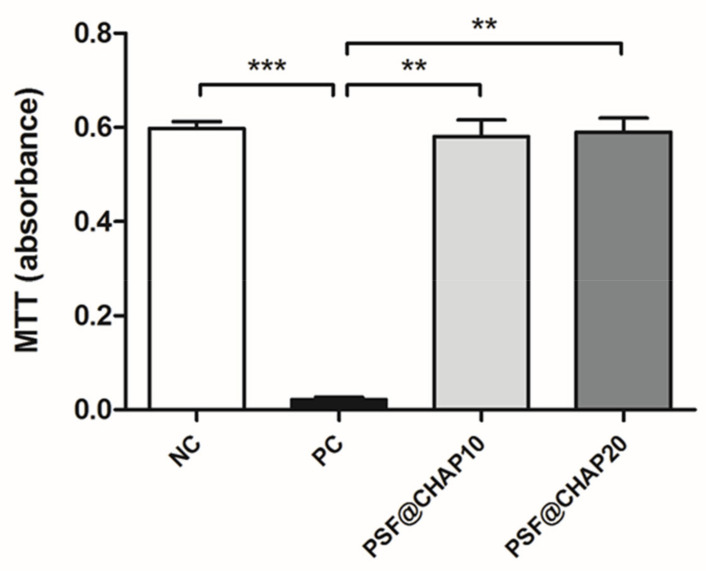
Cytotoxicity evaluation. Absorbance values of NIH-3T3 cells indirectly cultured with either PSF@CHAP10 or PSF@CHAP20 sponges, compared to cells cultured in fresh medium (negative control, NC) and in medium containing 0.1% TritonX-100 (positive control, PC). Data are reported as absorbance values; ** *p* < 0.01 and *** *p* < 0.001.

**Figure 5 nanomaterials-12-03182-f005:**
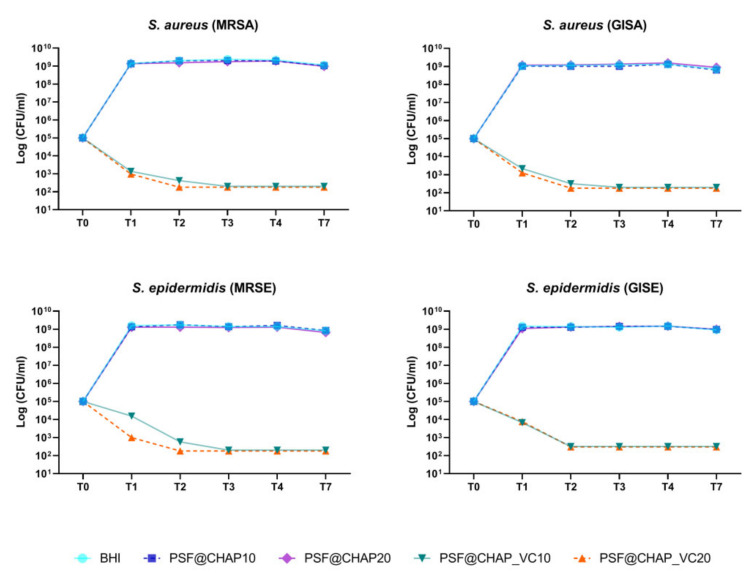
Bactericidal activity of PSF@CHAP_VC10 and PSF@CHAP_VC20 sponges against staphylococcal clinical isolates over time. Data are reported as log_10_ (CFU/mL).

**Figure 6 nanomaterials-12-03182-f006:**
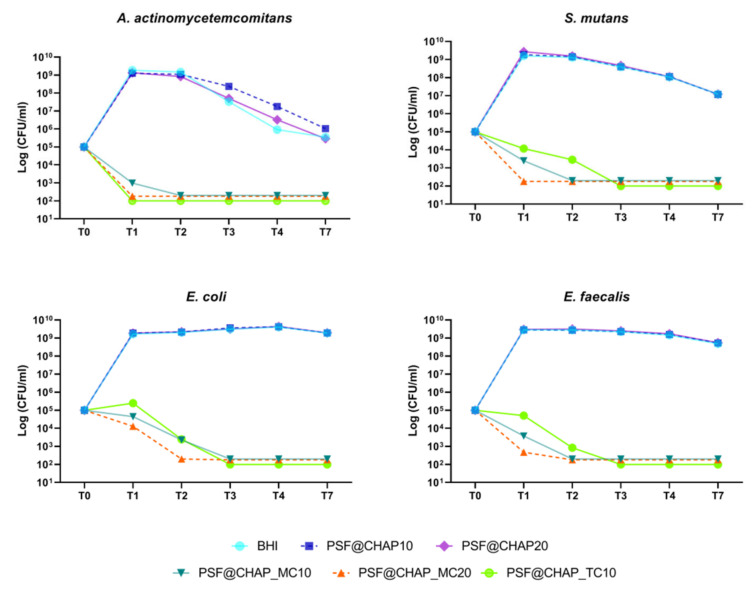
Bactericidal activity of PSF@CHAP_MC10, PSF@CHAP_MC20 and PSF@CHAP_TC10 against *A. actinomycetemcomitans*, *S. mutans*, *E.coli* and *E. faecalis*. Data are reported as log_10_ (CFU/mL).

**Figure 7 nanomaterials-12-03182-f007:**
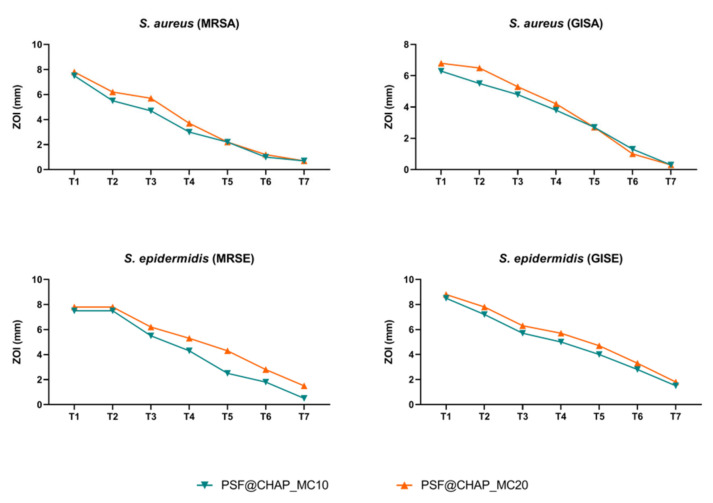
Zone of inhibition (ZOI) of MRSA, GISA, MRSE and GISE, exposed to PSF@CHAP_VC10 or PSF@CHAP_VC20 sponges over time. ZOIs are reported as mm ± SD and are plotted against time (T0 to T7).

**Figure 8 nanomaterials-12-03182-f008:**
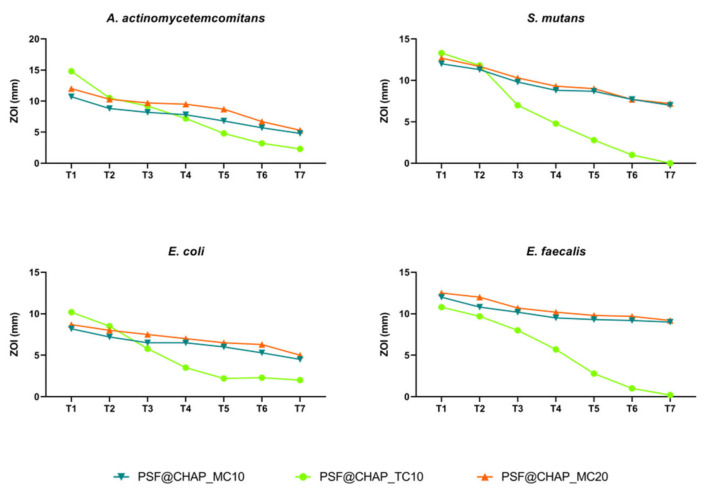
Zone of inhibition (ZOI) of *A. actinomycetemcomitans*, *S. mutans*, *E. coli* and *E. faecalis* exposed to PSF@CHAP_MC10, PSF@CHAP_MC20 and PSF@CHAP_TC10 sponges over time. ZOIs are reported as mm ± SD and are plotted against time (T0 to T7).

**Table 1 nanomaterials-12-03182-t001:** Resistance profile of the tested staphylococci to vancomycin before (T0); after a 7-day exposure to either PSF@CHAP_VC10 or PSF@CHAP_VC20 sponges (T7); and after a 7-day culture on antibiotic-free TSA (T14). MIC values are expressed as μg/mL.

	T0	Antibiotic Exposure	T7	T14
MRSA	0.125–0.25	PSF@CHAP_VC10	0.125–0.25	0.125–0.25
		PSF@CHAP_VC20	0.125–0.25	0.125–0.25
GISA	1	PSF@CHAP_VC10	1	1
		PSF@CHAP_VC20	1	1
MRSE	0.25	PSF@CHAP_VC10	0.125–0.25	0.125
		PSF@CHAP_VC20	0.125–0.25	0.125–0.5
GISE	1	PSF@CHAP_VC10	1	1
		PSF@CHAP_VC20	1	1

**Table 2 nanomaterials-12-03182-t002:** Resistance profile of the tested bacteria to minocycline and tetracycline before (T0), after a 7-day exposure to either PSF@CHAP_MC10, PSF@CHAP_MC20 or PSF@CHAP_TC10 sponges (T7), and after a 7-day culture on antibiotic-free TSA (T14). MIC values are expressed as μg/mL.

	T0	Antibiotic Exposure	T7	T14
*A. actinomycetemcomitans*	0.03	PSF@CHAP_MC10	0.03	0.03–0.06
		PSF@CHAP_MC20	0.03	0.03–0.06
*S. mutans*	0.06	PSF@CHAP_MC10	0.06	0.06
		PSF@CHAP_MC20	0.06	0.06
*E. coli*	0.5	PSF@CHAP_MC10	0.5	0.5
		PSF@CHAP_MC20	0.5	0.5
*E. faecalis*	0.03	PSF@CHAP_MC10	0.03	0.03–0.06
		PSF@CHAP_MC20	0.03	0.03–0.06
*A. actinomycetemcomitans*	0.25	PSF@CHAP_TC10	0.25	0.25–0.5
*S. mutans*	0.25	PSF@CHAP_TC10	0.25	0.25
*E. coli*	1	PSF@CHAP_TC10	1–2	1
*E. faecalis*	0.5	PSF@CHAP_TC10	0.5	0.5

## Data Availability

Not applicable.

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
