# Peer review of "Antibiotic-Loaded Hyperbranched Polyester Embedded into Peptide-Enriched Silk Fibroin for the Treatment of Orthopedic or Dental Infections"

_nanomaterials, 2022, doi:10.3390/nano12183182_

Round 1

Reviewer 1 Report

The article entitled “Antibiotic-loaded hyperbranched polyester embedded into Peptide-enriched Silk Fibroin for the treatment of orthopedic or dental infections” is submitted by Zili Sideratou, Marco Biagiotti, Dimitris Tsiourvas, Katerina N. Panagiotaki, Marta V. Zucca, Giuliano Freddi, Arianna B. Lovati, and Marta Bottagisio for publication in the journal Nanomaterials (MDPI).

This research deals with the synthesis and characterization of polymer nanoparticles integrated into peptide-enriched silk fibroin. This innovative material is expected to be used for orthopedic and dental applications. Specifically, the material is loaded with antibiotics whose release in the physiological environment can be controlled.

The physicochemical characterizations are carried out by FTIR, UV, and NMR spectroscopies, DLS, and zeta potential measurements. Antibiotic release, cell toxicity, and antibacterial activity are evaluated in a physiological environment.

The study contains scientifically interesting results. The text is very clear and thoroughly written. The discussion provides a lot of information and an appropriate comparison with the scientific literature on the topic to highlight the novelty of this research and the great interest for the expected biomedical applications.

In my opinion, this article can be accepted for publication in the journal Nanomaterials (MDPI) after modifying the following minor points:

 - in the introduction, line 40, it seems not appropriate to indicate that titanium is a bioinert metal, whereas it is indicated in line 41 that one of the main advantages of titanium is osseointegration properties. From my point of view, "bioinert" means that nothing happens with the biological environment. I think that is not true. I suggest removing this word.

 - In line 179, maybe one sentence can be added to explain what the CONTIN algorithm is. I think that is not obvious for all the readers.

 - Equation (1), why is the ratio multiplied by 100? If it is to get a percentage, that is not necessary to multiply the ratio.

 - everywhere in the text (lines 181, 365, 366, 368, 369, …), the zeta potential must be written with the Greek letter ζ (as done in line 171), or with the word "zeta".

Author Response

The authors would like to thank the reviewer for appreciating our manuscript and for all the useful and helpful comments. All comments have been taken into account and the paper has been revised accordingly.

Point 1: In the introduction, line 40, it seems not appropriate to indicate that titanium is a bioinert metal, whereas it is indicated in line 41 that one of the main advantages of titanium is osseointegration properties. From my point of view, "bioinert" means that nothing happens with the biological environment. I think that is not true. I suggest removing this word.

Response 1: The authors agree with the Reviewer’s suggestion. Accordingly, we removed the “bioinert” word from the text.

Point 2: In line 179, maybe one sentence can be added to explain what the CONTIN algorithm is. I think that is not obvious for all the readers.

Response 2: We thank the Reviewer for highlighting the not obvious meaning of the CONTIN algorithm. So, as suggested, a short text explaining the CONTIN algorithm is inserted in the 2.4 section, last paragraph and a relevant new reference is added. Specifically, the relevant text is: “For each sample, ten measurements were collected and the autocorrelation function obtained was analyzed employing the CONTIN algorithm [37], which is ideal for both monodisperse and polydisperse systems.”

Point 3: Equation (1), why is the ratio multiplied by 100? If it is to get a percentage, that is not necessary to multiply the ratio.

Response 3: If we do not multiply the Dt / Dtot ratio, the resulting values will be less than 1. Thus, in order to get a percentage, it is necessary to multiply the ratio by 100. To be more clear, the equation (1) is modified by inserting the Dt / Dtot ratio in parenthesis:

 drug release (%) = (Dt / Dtot) x 100

Point 4: everywhere in the text (lines 181, 365, 366, 368, 369, …), the zeta potential must be written with the Greek letter ζ (as done in line 171), or with the word "zeta".

Response 4: Based on the Reviewer’s request, the authors modified z- or ζ-potential as “zeta-potential” along the entire manuscript.

Reviewer 2 Report

This is an interesting study about antibiotic-loaded hyperbranched polyester embedded into Peptide-enriched silk fibroin for the treatment of orthopedic or dental infections. I recommend it for publication after the following minor issues are addressed.

1. In the introduction, the authors should add more discussion about why hyperbranched polyester is preferred.

2. It is not clear only partial OH groups of BH40 were chosen to be modified.

3. Several recent studies (doi.org/10.1016/j.actbio.2022.05.020; 10.1021/acs.langmuir.1c01312) related to silk fibroin materials should be included.

Author Response

The authors would like to thank the reviewer for considering our manuscript interesting. All the useful comments have been taken into account to ameliorate the quality of our study.

Point 1: In the introduction, the authors should add more discussion about why hyperbranched polyester is preferred.

Response 1: Following the Reviewer’s suggestion, the last paragraph of the introduction was expanded to clearly justify why hyperbranched polyester is preferred, as follows: “This polyester is synthesized in the molten state using 2,2-bis(hydroxylmethyl)propane-1,3-diol and 2,2- bis(hydroxymethyl) propionoic acid [35]. Due to its lipophilic interior, it was used to encapsulate lipophilic molecules, while it was found to be bio-degradable since enzymatic degradation of its ester bonds was observed [33]. Being water-insoluble, it is advantageous to be appropriately functionalized with hydrophilic end groups (e.g. carboxylates or polyethylene glycol moieties), thus resulting in micellar-type nanosystems [33, 34]. Additionally, after the initial screening of a number of hyperbranched polymers, the carboxylated derivative was found to be compatible with PSF solutions”.

Point 2: It is not clear only partial OH groups of BH40 were chosen to be modified.

Response 2: Section 3.1 has been modified in order to become clear that partial functionalization of the OH end groups of BH40 was achieved, as follows: “The partial functionalization of the commercially available hyperbranched aliphatic polyester Boltron H40TM (BH40) was achieved by the reaction of the hydroxyl end groups of BH40 (44 OH groups per molecule [33]) with succinic anhydride in anhydrous basic conditions (pyridine)”.

Point 3: Several recent studies (doi.org/10.1016/j.actbio.2022.05.020; 10.1021/acs.langmuir.1c01312) related to silk fibroin materials should be included.

Response 3: The authors appreciate the Reviewer’s suggestion to include more complete scientific literature related to silk fibroin materials. For this reason, we included in the text and in the reference list the following citations:

  1. Gomes, J. M.; Silva, S. S.; Fernandes, E. M.; Lobo, F.C.M.; Martín-Pastor, M.; Taboada, P.; Reis, R.L. Silk fibroin/cholinium gallate-based architectures as therapeutic tools. Acta Biomaterialia. 2022, 147, 168-184. doi: 10.1016/j.actbio.2022.05.020.
  2. Aye, S.S.S.; Zhang, Z-H.; Yu, X.; Yu, H.; Ma, W-D.; Yang, K.; Liu, X.; Li, J.; Li, J-L. Silk Hydrogel Electrostatically Functionalized with a Polycationic Antimicrobial Peptide: Molecular Interactions, Gel Properties, and Antimicrobial Activity. Langmuir. 2022, 38, 50-61. doi: 10.1021/acs.langmuir.1c01312
